# Programmable Physical Properties of Freestanding Chitosan Membranes Electrofabricated in Microfluidics

**DOI:** 10.3390/membranes13030294

**Published:** 2023-02-28

**Authors:** Khanh L. Ly, Piao Hu, Christopher B. Raub, Xiaolong Luo

**Affiliations:** 1Department of Biomedical Engineering, School of Engineering, Catholic University of America, Washington, DC 20064, USA; 2Department of Mechanical Engineering, School of Engineering, Catholic University of America, Washington, DC 20064, USA

**Keywords:** chitosan membrane, microfluidics, electrofabrication, programmable physical properties

## Abstract

Microfluidic-integrated freestanding membranes with suitable biocompatibility and tunable physicochemical properties are in high demand for a wide range of life science and biological studies. However, there is a lack of facile and rapid methods to integrate such versatile membranes into microfluidics. A recently invented interfacial electrofabrication of chitosan membranes offers an in-situ membrane integration strategy that is flexible, controllable, simple, and biologically friendly. In this follow-up study, we explored the ability to program the physical properties of these chitosan membranes by varying the electrofabrication conditions (e.g., applied voltage and pH of alginate). We found a strong association between membrane growth rate, properties, and fabrication parameters: high electrical stimuli and pH of alginate resulted in high optical retardance and low permeability, and vice versa. This suggests that the molecular alignment and density of electrofabricated chitosan membranes could be actively tailored according to application needs. Lastly, we demonstrated that this interfacial electrofabrication could easily be expanded to produce chitosan membrane arrays with higher uniformity than the previously well-established flow assembly method. This study demonstrates the tunability of the electrofabricated membranes’ properties and functionality, thus expanding the utility of such membranes for broader applications in the future.

## 1. Introduction

Membrane technology is crucial to selectively transport substances of interest in fluidic networks for a wide range of applications, including dialysis [1], extraction/separation [2,3], filtration [4], chemical gradient generation [5], and gas–liquid exchange [6], among others. With the boom of microfluidics technology since the 1990s, the integration of membranes and microfluidics has a growing interest among academic and industrial communities [7]. Membrane-integrated microfluidic platforms are commonly used in analytical biochemistry owing to their small reagent consumption, quick reaction time, cost-efficiency, high resolution, and controllability as compared to traditional technologies [8,9]. The utilization of membrane-integrated microfluidics for life science studies (e.g., cellular biology, biomedicine, and tissue engineering) demands membranes with enhanced biological compatibility and functionality [10]. Membranes derived from biocompatible materials have been used as scaffolds for cell, tissue, and organ-on-a-chip culture [11,12,13,14,15]. To better mimic the complexity and functionality of biology, a programmable yet facile membrane–microfluidic integration method is highly desired.

Currently available membrane–microfluidic integration approaches include (i) direct incorporation of commercially available membranes into a microchip, (ii) one-step preparation of a membrane-integrated microdevice, and (iii) in-situ preparation of membranes in a microdevice [7]. Despite extensive usefulness, existing membrane–microfluidic integration approaches possess certain drawbacks. First, improper and complicated membrane sealing in the direct incorporation approach can lead to leakage that induces device malfunction, such as toxicity to biological entities in the microfluidic chambers [16]. Second, the method to prepare membrane-integrated microdevices in one step is often material-dependent, complicated, and costly [7,17]. That being said, the in-situ fabrication of membranes in a microdevice, which readily forms membranes within microfluidic networks via polymerization reactions of monomers or biopolymers, can be a promising alternative. For instance, a porous filter made of 2-hydroxyethyl methacrylate monomers was fabricated in situ within microchannels using emulsion photopolymerization [18]. Later, Nair and colleagues presented an approach to synthesize bovine albumin membranes in situ using an interfacial crosslinking reaction [19]. Nevertheless, the residual crosslinker after membrane formation in these approaches might induce cellular toxicity, limiting the utility of those membranes for biological applications.

To tackle unwanted toxicity, Luo et al. demonstrated a facile in-situ biofabrication of flow-assembled chitosan membrane arrays in a single-layer polydimethylsiloxane (PDMS) device without a toxic crosslinker [20]. Chitosan has long been recognized for its biocompatibility, biofunctionality, controlled biodegradation, and flexible modification of functional groups that promote its utility in agriculture, biological, biomedical, and tissue engineering [21,22,23]. Importantly, the pH-dependent sol–gel transition of chitosan-based on amine chemistry (Figure 1A) enables a facile and reversible biofabrication of chitosan membrane-integrated microchips by simply manipulating the pH of the surrounding aqueous environment [20,24]. Using this amine chemistry of chitosan, Luo’s group integrated chitosan membranes in microfluidic platforms with flows by creating localized pH gradients at the flow interface for numerous applications ranging from chemical gradient generation [20], chemotaxis [25], and chemotropism [26] to cellular signaling and communications [27,28,29]. On the other hand, Payne’s group has pioneered the use of electrical stimuli to form spatiotemporally programmable chitosan membranes on the cathode surface, which have been used as bioelectronic sensors [30,31], protective barriers against redox reactions [32], and antimicrobial films [33]. Indeed, the use of electrical stimuli to guide chitosan membrane formation offers not only biocompatibility but also facile manipulation over experimental parameters such as current, voltage, or time with a high level of controllability [34].

Recently, Luo’s group presented a novel interfacial electrofabrication of chitosan membranes with distal electrodes [35]. Compared to previous cathode-based electrofabrication, the new interfacial electrofabrication offers several unique advantages. First, the process is flexible, and the location of membrane formation is controllable without the need for electrodes at the membrane fabrication site. Second, metallic ion contaminations can be limited since no metallic cathode at the fabrication site is needed locally. Finally, the membrane freestanding configuration allows for unlimited electrical and fluidic accesses to both sides of the electrofabricated membranes, presenting the versatile applicability of such membranes in biochemistry, sensors, or energy storage [35]. The electrodeposition of chitosan membrane/hydrogel on the electrode surface has long been recognized for its spatiotemporal programmability and incorporation of biological entities [36,37,38,39,40]. We hypothesize that the chitosan membranes electrofabricated using this newly invented method can also be precisely controlled and should be further explored in terms of spatiotemporal resolution, physical properties, and biological functionalization. For example, understanding the association between the chitosan membrane’s properties and the fabrication conditions is important to actively manipulate the properties of the chitosan membrane-integrated microfluidic platform for diverse applications.

In this work, we aim to study the programmability of the physical properties of chitosan membranes interfacially electrofabricated in a microfluidic network by varying the fabrication parameters (e.g., voltage and pH values of alginate). The effects of fabrication parameters on the resulting membranes were evaluated through the membrane growth rate, the membrane porosity (or permeability), and the microstructural alignment and orientation of chitosan chains within the membranes. We also demonstrated the ability to scale up this interfacial electrofabrication technique for producing chitosan membrane arrays compared to a well-established flow assembly approach. The study enhances our understanding of the properties and functionalities of these electrofabricated membranes and allows for active tailoring of the membrane according to application needs.

## 2. Materials and Methods

### 2.1. Materials

Alginate solution of 0.5% *w*/*v* (from sodium alginate powder with medium viscosity, Sigma-Aldrich, St. Louis, MO, USA) and chitosan solution of 0.5% *w*/*v* (from 85% deacetylated chitosan flakes with medium molecular weight, Sigma-Aldrich, St. Louis, MO, USA) were prepared as reported [41]. The pH of the alginate solution was adjusted to 6, 8, 10, and 11.5, while the pH of the chitosan solution was maintained at around 5.5 for the electrofabrication of chitosan membranes. Fluorescein isothiocyanate-labeled dextran (F-dextran) with molecular weights of 4 kDa and 1× phosphate-buffered saline (PBS) were obtained from Sigma-Aldrich (St. Louis, MO, USA). Other chemicals used in this study can be obtained from major suppliers.

### 2.2. Microfluidic Platform Preparation

The SU-8 mold on a 4” silicon wafer with single apertures and three-channel micropatterns used in this study was fabricated using the photolithography method as reported [42]. PDMS microchannels were fabricated with the soft lithography technique using the patterned SU-8 mold. The solidified PDMS was peeled off the mold and cut into a set of 4–5 microchannels, punched for inputs and outputs with a 22 ga biopsy puncher. The PDMS microchannels were then bounded to cleaned glass slides using oxygen plasma treatment with Plasma Cleaner PDC-32G (Harrick Plasma, Ithaca, NY, USA) to form a complete PDMS microfluidic device [43]. Two types of PDMS devices were used. The first type was a single-aperture PDMS device composed of two microchannels, as depicted in Figure 1B. The two channels of 500 μm × 50 μm (W × H) were connected via a 50 μm wide aperture (Figure 1C(i)). The second type was a three-channel PDMS device composed of three microchannels, as shown in Figure 8B. The three microchannels of 500 µm × 50 µm (W × H) were separated by two arrays of six 100 µm × 50 µm (L × W) PDMS pillars and connected through two arrays of seven 50 µm × 50 µm square apertures (Figure 8A,B).

### 2.3. Fabrication of Chitosan Membranes

The interfacial electrofabrication of chitosan membrane(s) was conducted following previous works with modifications [35,44,45]. Two (for single-aperture PDMS device) or three (for three-channel PDMS device) metal couplers (22 ga × 0.3 inch, Instech Laboratories, Inc., Plymouth Meeting, PA, USA) were used as both capillary connectors and distal electrodes for the interfacial electrofabrication. They were inserted into the input of each channel while the output was left open. To fabricate membrane arrays in the three-channel device, the two metal couplers in alginate channels were wired to establish a complete electrical circuit. In both devices, the chitosan channel was connected to the anode, and the alginate channel/channels were connected to the cathode. After chitosan and alginate solutions were introduced into the channels using syringe pumps (NE-1000, New Era Pump Systems, Inc., Farmingdale, NY, USA) at the flow rate of 0.8 μL/min, an air bubble(s) was trapped within the aperture(s) due to the hydrophobic nature of PDMS, as shown in Figure 1C(i) (and Figure 8B(i)). The flows were stopped, and the air bubble(s) was dissipated with an add-on vacuum chamber, as shown in Figure 1B [46]. A polyelectrolyte complex membrane (PECM) instantaneously formed due to the electrostatic interaction between positively charged chitosan and negatively charged alginate chains (Figure 1C(ii)) [47]. Next, a direct voltage, varying from 1.5 V to 9 V with a 1.5 V increment stepwise, from a direct power supply (2450 Keithley Source Meter, Keithley Instruments, Cleveland, OH, USA) was applied, and a chitosan membrane grew on the PECM over time (Figure 1C(iii,iv)). All membranes were electrofabricated in 5 min while the flows were in stop condition unless specified otherwise. On the other hand, the flow-assembled chitosan membrane was fabricated in a similar way as the above procedure, except (1) no electrical signal was applied, and (2) the flows were restarted once PECM(s) was formed to generate a localized pH gradient at the flow interfaces to drive the membrane formation. Afterward, the microchannels with formed chitosan membranes were manually rinsed with PBS and stored at 4 °C for further analysis.

To better demonstrate the formation of chitosan membranes in electrofabrication and flow assembly, fluorescently labeled polystyrene beads (FluoSpheres^®^ Fluorescent Microspheres, 200 nm diameter beads, Thermo Fisher Scientific, Waltham, MA, USA) were mixed with chitosan solution at the ratio of 1:20 to visualize the deposition of chitosan chains during membrane formation, as discussed in the Results section.

The effects of fabrication parameters on the chitosan membrane’s properties were investigated by varying the applied voltage and the pH of the alginate solution. Table 1 summarizes the samples being investigated with respect to the fabrication parameters.

### 2.4. Membrane Growth Rate

During the fabrication of chitosan membranes, optical images were taken every 30 s, and the membrane thickness was measured with ImageJ 1.51j8 (NIH, Bethesda, MD, USA). The growth rate of chitosan membranes was determined by plotting the membrane thickness with respect to time [46,48], which revealed the correlation between membrane growth and fabrication parameters (voltage and pH of alginate).

### 2.5. Permeability Test

To investigate the effects of fabrication parameters on the semi-permeability of chitosan membranes, the F-dextran transport experiment was conducted in the single-aperture, chitosan membrane-integrated PDMS device as reported [49]. In brief, 1 mg/mL F-dextran solution was continuously supplied to the channel on the PECM side at the flow rate of 1 µL/min, while the other channel was filled with PBS in stop condition. The fluorescence images were taken after 5 min. Fluorescence intensity in the solution close to the PECM side (source) and close to the chitosan membrane side (sink) was quantified with ImageJ, as indicated in Figure 4A. The semi-permeability of chitosan membranes was quantified as the percentage of F-dextran penetrated through the membrane, which was calculated by dividing the fluorescence intensity of the sink by that of the source.

### 2.6. Birefringence and Parallelism Index Analysis

The birefringence signal and parallelism index (PI) were used to assess the microstructural organization of chitosan chains within the fabricated chitosan membranes. To quantify birefringence, the optical retardance of light passed through chitosan membranes was measured based on the de Sénarmont compensation technique as reported [41,49,50,51]. The microfluidic device containing fabricated chitosan membranes was imaged with a quantitative polarized microscope (qPLM, MT9930, Meiji Techno America, San Jose, CA, USA). First, the qPLM was calibrated with the polarizer and analyzer angle set to 0° and 90°, respectively, measured on a Vernier scale to 0.1° accuracy and confirmed to be at maximum extinction of the background. Then, a de Sénarmont compensator plate and a green filter (547 ± 20 nm) were inserted above the linear analyzer and beneath the polarizer, respectively. Subsequently, the microfluidic device containing chitosan membranes was rotated on the imaging stage until the highest birefringence signal around the central region of the membrane was observed. Then, the sample stage was locked, and the analyzer was rotated counterclockwise while images were taken every 1° until all portions of the chitosan membrane passed through a minimum pixel value. The optical retardance was quantified from the birefringence signal for each pixel of the images by fitting the pixel signal with respect to the analyzer angle to a second-order polynomial, determining the minimum, and finally generating an optical retardance map of the membrane [50].

On the other hand, the PI was calculated as follows:$$PI=\frac{I_{max}-I_{min}}{I_{max}+I_{min}}$$
where $I_{max}$ and $I_{min}$ were determined for each pixel by removing the compensator from the light path, rotating the polarizer and analyzer in 15° increments over 90°, and fitting the signal (*I*) with the rotation angle (*θ*) to the function
$$I=A+B\;\sin^{2}(2\;\theta +\varphi )$$
where *A*, *B*, and φ are parameters determined from a nonlinear least square fit [52,53].

### 2.7. Statistical Analysis

All experiments were conducted with sample size n = 3–4 chitosan membranes per fabrication condition. Data are expressed as mean ± standard deviation (SD). The effects of time and voltage or pH of alginate on membrane’s growth rate and the effects of voltage and pH of alginate on chitosan membrane’s permeability and optical retardance were evaluated using two-factor ANOVAs (SPSS, v28.0, SPSS Inc., Chicago, IL, USA). Then, if any factor (voltage or pH of alginate) was significant, one-factor ANOVAs were performed with Tukey’s post hoc test to compare between groups at each time point. The level of significance was set at 0.05. Membrane thickness, semi-permeability, and retardance were normalized with the maximum value of 1. Linear regression analysis was performed on normalized data to investigate the dependence of (1) membrane thickness, (2) membrane semi-permeability, and (3) membrane retardance on voltage and pH. Pearson’s correlation test was performed to determine the association of the membrane’s permeability versus its retardance within electrofabricated membranes.

## 3. Results

### 3.1. Membrane Growth Rate

First, we determined the voltage boundaries feasible for the interfacial electrofabrication of chitosan membranes by varying the voltage (1.5–9 V) while fixing the pH of the alginate solution (pH = 8). Figure 2A–C shows the representative fabricated membranes using alginate solution at pH 8 while increasing the voltage. It is clear that the higher the voltage, the thicker the membrane. Quantitatively, Figure 2D shows that the membrane thickness depended on time (ANOVA, F = 222.1, *p* < 0.001), voltage (ANOVA, F = 571.1, *p* < 0.001), and their interactions (ANOVA, F = 14.4, *p* < 0.001). After 5 min of fabrication, the chitosan membrane reached 8.6 ± 4.7, 32.4 ± 3.1, 40.1 ± 1.5, 48.9 ± 1.5, and 69.1 ± 11.2 µm with respect to the voltage of 1.5, 3, 4.5, 6, and 7.5 V, respectively. Noticeably, the lowest voltage source (1.5 V) could only form a relatively thin membrane under 10 µm, not mechanically robust enough for most applications, while the highest voltage source (7.5 V) produced relatively inconsistent membranes with a large batch-to-batch variation. Figure 2E shows the percentages of coefficient of variation of the five tested groups, in which the values for membranes electrofabricated under 1.5 and 7.5 V were dramatically large at 54.8% and 16.2%, respectively. Meanwhile, membranes electrofabricated using 3–6 V yielded a relatively small coefficient of variation under 10%. Thus, the voltage range of 3–6 V was deemed feasible for interfacial electrofabrication in this study.

Next, we investigated the dependency of membrane growth on the pH of the alginate solution. Figure 3 shows the growth rates of electrofabricated chitosan membranes under varied voltage and pH of alginate. In general, the growth rates of membranes show an obvious pattern in which faster growth was associated with a higher voltage applied with statistical significance. Specifically, at a fixed alginate pH of 6, the membrane thickness depended on time (ANOVA, F = 502.0, *p* < 0.001), voltage (ANOVA, F = 396.0, *p* < 0.001), and their interactions (ANOVA, F = 6.3, *p* < 0.001). At a fixed alginate pH of 10, the membrane growth depended on time (ANOVA, F = 605.3, *p* < 0.001), voltage (ANOVA, F = 296.6, *p* < 0.001), and their interactions (ANOVA, F = 5.6, *p* < 0.001). At a fixed alginate pH of 11.5, the membrane growth depended on time (ANOVA, F = 485.8, *p* < 0.001), voltage (ANOVA, F = 458.3, *p* < 0.001), and their interactions (ANOVA, F = 7.4, *p* < 0.001). The differences in membrane thickness among different voltages were already observed within the first 30–60 s of fabrication. The final thickness of the resulting membranes is plotted in Figure 3D. Statistically, the final membrane thickness depended on the applied voltage (ANOVA, F = 31.4, *p* < 0.001), pH of the alginate solution (ANOVA, F = 169.3, *p* < 0.001), and their interactions (ANOVA, F = 3.0, *p* < 0.05). The membranes electrofabricated using 6 V yielded the highest thickness, while those fabricated using 3 V were the thinnest regardless of the pH of alginate solution. To better demonstrate the pH dependency of membrane growth rate, we plotted the membrane growth rates of varied pH of the alginate at a fixed voltage (Figure 3E). At the fixed voltage of 4.5 V, the membrane thickness depended on time (ANOVA, F = 849.7, *p* < 0.001), pH of alginate (ANOVA, F = 174.8, *p* < 0.001), and their interactions (ANOVA, F = 4.2, *p* < 0.001). In particular, at low pH values of 6 and 8, the differences in the membrane growth rates were not significant, with the final membrane thickness being 38.5 ± 0.5 µm and 39.5 ± 1.2 µm, respectively. As the pH of alginate was increased to 10 or 11.5, the difference in membrane growth rate was more significant: the higher the pH, the faster the growth. The membrane yielded the highest thickness of 58.1 ± 1.2 µm when the voltage of 6 V and the pH 11.5 of alginate were used.

### 3.2. Membrane Permeability

The permeability test was conducted as described in Section 5 for 30 min, qualitatively monitored, and quantitatively calculated as the percentage of F-dextran (MW 4 kDa) penetrating through the membrane by dividing the fluorescence intensity at the sink by that at the source, as indicated in Figure 4A. Since the permeability of the membrane depends not only on the porosity but also on the permeation of traveling molecules, we purposely prepared a separate batch of membranes of the same thickness (~30 µm) regardless of fabrication conditions for the permeability test. Figure 4B shows the zoomed-in images of different electrofabricated chitosan membranes during the permeability test. The membrane permeability can be qualitatively observed by examining the fluorescence signal within the chitosan membranes. With the fixed voltage at 3 V, the highest transportation of F-dextran was associated with the lowest pH at 6 (Figure 4B(i)). Noticeably, a much lower amount of F-dextran was able to pass through the membrane electrofabricated using pH 11.5 alginate and 6 V voltage as compared to those electrofabricated at 3 or 4.5 V or flow-assembled (Figure 4B(ii)). Figure 4C shows the quantified percentages of F-dextran between the sink and source areas, as indicated in Figure 4A. Overall, the permeability of chitosan membranes varied with the applied voltage (ANOVA, F = 283.4, *p* < 0.001) and pH of alginate solution (ANOVA, F = 16.8, *p* < 0.001), and the interactions between voltage and pH of alginate (ANOVA, F = 2.6, *p* < 0.05). With the fixed voltage at 3 V, the membrane permeability increased from 7.3 ± 0.2% at pH 11.5 to 12.0 ± 1.5% at pH 6, or a 1.6× increase in permeability. Meanwhile, with the fixed pH at 11.5, the membrane permeability increased from 0.2 ± 0.2% at 6 V to 7.3 ± 0.2% at 3 V, or a 36.5× increase in permeability. The highest F-dextran permeability was observed through pH6_3V (12.0 ± 1.5%), while the lowest F-dextran permeability belongs to pH11.5_6V (0.2 ± 0.2%).

Interestingly, flow-assembled chitosan membranes yielded F-dextran permeability of 12.8 ± 0.3%, much higher than that of membranes electrofabricated using the same alginate solution at any applied voltage, ranging from 0.2 ± 0.2% to 7.3 ± 0.2%. This suggests that the interfacially electrofabricated membranes might possess much denser microstructure that limited the permeability of F-dextran as compared to the flow-assembled ones. Furthermore, the permeability of the electrofabricated chitosan membrane ranges broadly from 0.2 ± 0.2% at 6 V, pH 11.5 to 12.0 ± 1.5% at 3 V, pH 6, which is a 60× difference. This suggests that the electrofabricated chitosan membrane permeability is programmable by actively manipulating the fabrication conditions, including the applied voltage and pH of the alginate solution.

### 3.3. Membrane Birefringence

In previous work, optical retardance, a parameter representing birefringence signal, was used as an indicator of the molecular alignment of flow-assembled chitosan membranes [41,49,51]. In this work, we also employed this optical parameter to determine the effects of electrofabrication conditions on the microstructural organization of electrofabricated membranes. Figure 5A shows the typical optical retardance maps of electrofabricated membranes under varying conditions. Visually, the optical retardance signal increased with increasing the pH of the alginate solution at a fixed voltage of 3 V (Figure 5A(i)). Similarly, the higher the applied voltage, the higher the optical retardance obtained, as seen in Figure 5A(ii). The profiles of optical retardance from the PECM side towards the other end of the membrane edge, taken as a percentage of membrane thickness, were plotted in Figure 5B. It is clear that there was a steep decelerating profile of retardance from the PECM side to the chitosan side, and the higher the applied voltage, the steeper the profile. By analyzing the net optical retardance across the membrane, we found some interesting results. Overall, the optical retardance of the chitosan membrane depended on the applied voltage (ANOVA, F = 53.9, *p* < 0.001) and pH of alginate (ANOVA, F = 15.8, *p* < 0.001), and the interactions between voltage and pH of alginate (ANOVA, F = 3.5, *p* < 0.01). At a fixed voltage of 3 V, the pH-dependent birefringence was clearly observed, with the highest value of 24.9 ± 0.4 nm belonging to the membranes with a pH of 11.5. As the applied voltage increased, the net optical retardance of membranes also increased. For instance, at fixed alginate pH of 8 (red bars in Figure 5C), the retardance of membranes steadily risen from 15.8 ± 2 nm to 27.1 ± 3.7 nm and 31.7 ± 4.0 nm as the applied voltage increased from 3 V to 4.5 V and 6 V, respectively.

Next, we compared the physical properties of the electrofabricated chitosan membranes assembled using electrical stimuli with those flow-assembled to deepen the understanding of these membranes. The alginate solution with a pH of 11.5 was used for both the electrofabrication and flow assembly of chitosan membranes in this experiment. Further, the applied voltage was fixed at 4.5 V for electrofabrication. First, the molecular alignment of both electrofabricated and flow-assembled membranes was assessed through optical retardance and parallelism index. As seen in Figure 6A,B, the retardance signal of electrofabricated membranes was noticeably higher than that of the flow-assembled ones. In particular, the retardance profile was much stronger near the PECM side and with a steeply decreased slope towards the membrane’s edge in electrofabricated membranes. A consistent decreasing trend in retardance was also observed in flow-assembled membranes but at a less decreasing slope. Interestingly, although the PI profile of flow-assembled membranes was fairly stable around 0.86 ± 0.02, except for some fluctuation around the membrane’s edge (PECM and chitosan side), its profile was fairly stable around 0.86 ± 0.02. Meanwhile, the PI profile of electrofabricated membranes experienced a steady decline from the PECM side toward the chitosan side, similar to its retardance profile. The correlations between net optical retardance with PI were replotted in Figure 6C, which shows an obvious difference in optical retardance but little difference in PI (within error bars) between electrofabricated and flow-assembled chitosan membranes. Figure 6D clearly shows an inverse correlation between the optical retardance of chitosan membranes and their semi-permeability.

To further understand the growth mechanism and likely microstructural alignment of the chitosan membranes, we employed fluorescence-labeled particles to trace the trajectories and deposition of chitosan chains during the fabrication. Figure 7A,B shows the typical interfacial electrofabrication and flow assembly of chitosan membranes, respectively. It can be clearly seen that the chitosan chains were attracted vertically toward the PECM during electrofabrication. Concurrently, a flux of hydroxyl ions from the alginate solution was driven by electrical stimuli passed through the PECM to induce chitosan chain polymerization, resulting in insoluble chitosan membranes. We also performed the flow assembly of chitosan membranes with embedded fluorescence-labeled particles, as shown in Figure 7B. It clearly shows that the chitosan chains were assembled layer by layer in parallel with the PECM in the flow streamlines, where a localized pH gradient at the flow interface was generated by laminar flows to induce the sol–gel transition of the chitosan membrane on top of the PECM [54].

### 3.4. High Throughput Electrofabrication of Chitosan Membranes

In this section, we investigated the versatility of the interfacial electrofabrication in another microfluidic platform, a three-channel PDMS microdevice, as shown in Figure 8A [20]. In analogy, the fourteen apertures in the three-channel microdevice act as fourteen parallel resistors. By connecting the two alginate channels to a negative source and the chitosan channel to a positive source, we established a closed electrical circuit with fourteen parallel resistors, as depicted in Figure 8A. Chitosan membranes were formed in 10 min in this experiment for both interfacial electrofabrication and flow assembly approaches. 

Figure 8B shows the representative time-lapse images of two arrays, seven in each array, which were formed simultaneously in a three-channel device by electrofabrication. Similar to the previously reported flow assembly approach, we utilized the hydrophobicity of PDMS to spontaneously trap air bubbles within fourteen apertures (Figure 8B(i)). During the vacuuming of air bubbles, the flows in three channels were stopped, enabling stable solution interfaces to form PECMs (Figure 8B(ii)) [46]. Once PECMs were formed, the electrical signal was turned on to form chitosan membranes on top of PECMs (Figure 8B(iii,iv)). Figure 8B(iii) shows that the individual chitosan membranes were relatively uniform in the initial growth stage. As the chitosan membranes grew beyond the PDMS pillar corners and further into the chitosan channel (Figure 8B(iv)), they grew into semicircular shapes, presumably following the electric fields. The total diffusion length for small molecules around the PDMS pillar corners does not appear to be much different from those straight through the center of the membrane. Figure 8C(i) shows that the membrane growth rate strongly depended on the time (ANOVA, F = 279.5, *p* < 0.001), fabrication method (ANOVA, F = 1657.1, *p* < 0.001), and their interaction (ANOVA, F = 34.6, *p* < 0.001). From the growth curves, it is clear that the membranes’ growth in the two approaches was roughly comparable within the first 60 s of fabrication. Then, as the membranes grew thicker, the membrane growth in flow assembly slowed down and eventually reached a plateau, resulting in much thinner membranes. On the other hand, the growth of electrofabricated membranes did not slow down much and was nearly linear after the first 2–3 min [35]. After 10 min, the thickness of the electrofabricated membranes and flow-assembled membranes reached 50.1 ± 1.8 µm and 27.4 ± 5.3 µm, respectively. Interestingly, while the membrane thickness between the upstream versus downstream membranes by electrofabrication was similar, this was not the case for the flow-assembled membranes. Figure 8C(ii) shows that the upstream membranes were thicker while the downstream membranes were thinner, which was in agreement with our previous study due to the reduction in available chitosan molecules along the membrane growth front from upstream to downstream of flow [55]. Overall, the membrane thickness was more uniform by electrofabrication, suggesting that the electrofabrication approach is highly reproducible and spatially programmable.

## 4. Discussion

Programmable biofabrication of freestanding membranes in microfluidics is of particular interest that finds great utility for a wide range of biology and tissue engineering applications. In this study, we investigated the ability to program the physical properties of the interfacially electrofabricated chitosan membranes by controlling the applied voltage and pH of the alginate solution. We chose a constant voltage instead of constant current signals for electrofabrication in this study for several reasons. First, a constant voltage can be applied independent of aperture geometry, size, or the number of apertures. As shown in Figure 8, it was more convenient to use constant voltage when electrofabricating multiple membranes simultaneously. Second, since we intended to compare the properties of chitosan membranes formed with flow assembly and electrofabrication, the constant voltage scenario is more comparable to the flow assembly approach from the perspective of fixing the potential or pH gradient at the membrane growth front. The potential subtle difference between constant current and constant voltage will be investigated in the future.

Electrical stimuli have been employed to deposit a wide range of biopolymer coatings onto metallic surfaces [34,56,57,58]. Depositing chitosan onto the cathode surface via electrical signals has been widely explored by imposing a high pH gradient around the cathode [36,59]. The electrodeposition on the electrode surface, however, is not suitable for fabricating standalone membranes that allow for fluidic access to both sides of the structure for broader applications. Unlike the conventional approach, chitosan membranes in this study were electrofabricated on top of a freestanding, flexible, and thin PECM at the interface of chitosan and alginate solutions instead of cathode/metallic surfaces. A PECM was spontaneously formed due to the electrostatic interactions between positively charged amine groups of chitosan and negatively charged carboxyl groups of alginate chains. The PECM acted as a physical barrier to inhibit the mixing of chitosan and alginate macromolecules while allowing hydroxyl ions to diffuse across and induce the gelation of chitosan chains [10]. Besides the unique benefits mentioned above, the feasibility of manipulating PECM formation allows for precise yet flexible control over the location of the resulting membranes [35]. Compared to previous membrane-integrated platforms generally incorporated in vertical configuration and requiring extra steps for device packaging [60,61], our membrane integration is simple, and the membrane formation process is easy to track in real time using bright field microscopy. Further, the resulting membranes possess controllable thickness, tunable permeability, and programmable microstructure. The membranes are functional and stable in neutralized and moisture conditions for up to a month [20,35]. Finally, our freestanding configuration of the electrofabricated chitosan membranes within microfluidic networks differs from many claimed freestanding chitosan membranes in the literature cast in Petri dishes or electrodeposited onto electrodes and delaminated from hard surfaces [62,63,64].

Here, we explored the programmable physical properties of the interfacially electrofabricated chitosan membranes and obtained several key findings. First, the membrane growth was proportional to the increasing voltage and pH of alginate (Figure 2 and Figure 3). The voltage dependency agreed with previous studies where electrodeposited chitosan films on electrode surfaces increased with voltage [65]. Similarly, the elevated pH of alginate resulted in a faster growth rate and thicker membrane [35,41]. These could be explained by the higher flux of hydroxyl ions induced by higher voltage or higher solution pH that drove the membrane formation at a faster rate. Further, the higher level of molecular alignment of the resulting membranes, induced by the higher retardance, was associated with the higher applied voltage and pH of alginate and vice versa (Figure 5). The pH dependency of birefringence was consistent with previous work, and it could be explained by the higher flux of hydroxyl ions and steeper pH gradient that led to a higher net retardance and a steeper retardance profile decreasing from the PECM toward the other membrane edge. The voltage dependency of birefringence, however, was new in this study, which offered a simple way to manipulate the micro-alignment of the electrofabricated membranes by simply controlling the electrical signal. 

We also investigated the mathematical model of (1) membrane thickness, (2) membrane semi-permeability, and (3) membrane retardance as functions of voltage and pH using multiple linear regression analysis. Table 2 summarizes the multiple linear regression coefficients and R^2^ values of specific membrane properties with respect to pH and voltage. To fairly compare the regression coefficients, the thickness, retardance, and permeability columns were normalized so that each had a maximum value of 1. The equation of membrane thickness dependence on pH (t-statistic = 4.4 and *p* = 0.002) and voltage (t-statistic = 9.2 and *p* = 0.000) is:$$\;\mathrm{Predicted}\;\mathrm{membrane}\;\mathrm{thickness}=\left(0.07\pm 0.07\right)+\left(0.03\pm 0.01\right)\times \mathrm{pH}+\left(0.1\pm 0.01\right)\times \mathrm{voltage}$$

Meanwhile, the equation of membrane semi-permeability dependence on pH (t-statistic = −3.9 and *p* = 0.004) and voltage (t-statistic = −14.5 and *p* = 0.000) is: $$\mathrm{Predicted}\;\mathrm{membrane}\;\mathrm{semi}-\mathrm{permeability}=\left(1.83\pm 0.11\right)-\left(0.04\pm 0.01\right)\times \mathrm{pH}-\left(0.23\pm 0.02\right)\times \mathrm{voltage}$$

Finally, the equation of membrane retardance dependence on pH (t-statistic = 4.4 and *p* = 0.002) and voltage (t-statistic = 6.6 and *p* = 0.000) is: $$\mathrm{Predicted}\;\mathrm{membrane}\;\mathrm{retardance}=\left(-0.07\pm 0.11\right)+\left(0.04\pm 0.01\right)\times \mathrm{pH}+\left(0.11\pm 0.02\right)\times \mathrm{voltage}$$

The magnitudes of the slope coefficients were comparable, except the semi-permeability had a ~2-fold increased sensitivity to voltage compared to the other membrane properties. While the membrane thickness and retardance were positively proportional to the alginate pH and voltage, membrane semi-permeability was, in contrast, inversely proportional. This finding is consistent with the observation that higher retardance is generally produced from the denser and potentially more aligned chitosan membranes, with lower semi-permeability in denser membranes. Multiple R^2^ values of all three were over 0.93, indicating that the voltage and pH values during fabrication are sufficient to predict the developed membrane properties effectively.

The ability to actively tune the micro-alignment level of the resulting membrane also enabled the programmability of the membrane permeability or pore size. We found that the membrane permeability was inversely proportional to its optical retardance. The correlation between the membrane’s permeability and retardance among the electrofabricated membranes was assessed, and a linear relationship was observed as follows:y=32.54−1.28x
where *x* is the membrane permeability in percentage, and *y* is the membrane retardance in nm (Appendix A). The Pearson correlation test also suggested that the correlation between the two variables was significant with *p* = 0.001. This suggests that the higher retardance is associated with a denser membrane microstructure and smaller pore size, which eventually limits the diffusion of substances across the membrane and results in lower membrane permeability.

Moreover, there is a possible association between the membrane’s growth rate and the pore size or permeability of the membrane: the faster the membrane growth, the lower the permeability (Figure 3 and Figure 4). It is likely that a higher applied voltage and pH of alginate induced higher momentum onto the membrane growth front or stronger compression toward the PECM. Concurrently, the higher flux of hydroxyl ions driven by either higher applied voltage or pH of alginate induced a higher degree of deprotonation of the amine groups on chitosan chains or quicker polymerization of the chitosan chains without relaxation, thus resulting in greater molecular density in the membrane. By either mechanism, the ability to program the microstructural alignment and density of electrofabricated chitosan membranes with experimentally controllable parameters enables a promising membrane biofabrication with tailorable membrane microstructure and functionality, highly desirable and useful for biomedical and tissue-on-a-chip research. Besides altering the pH of alginate solution and electrofabrication voltage, it is also feasible to tune the membranes’ physicochemical properties with salt [66], crosslinking agents [51], nanoparticles [49], or by varying the concentration of chitosan [14]. Finally, the ability to electrofabricate chitosan membrane arrays with high uniformity as compared to the flow assembly method (Figure 8) is also interesting and should be further explored in future studies.

Noticeably, the optical retardance of the electrofabricated membranes was higher than that of the flow-assembled ones. A higher concentration of chitosan chains and the more aligned chitosan chains could both contribute to a higher optical retardance through form and intrinsic birefringence, respectively [67]. Presumably, there is a force normal to the electrofabricated membrane interface that exceeds the viscous drag on soluble chitosan chains, leading to a momentum transfer to the membrane when the soluble chain collides with the interface. This momentum transfer may lead to the efficient packing of chitosan chains in the membrane and contribute to the co-alignment of chains in the electrofabricated membrane. In the flow assembly case, the mechanism of chitosan chain packing and alignment is different, and presumably related to shear stress, which is tangential to the growing membrane interface. Further experiments and molecular simulations that model these different force environments at the chitosan membrane interface while parameterizing or quantifying local chitosan chain density and alignment would provide more detailed knowledge of chitosan chain organization within membranes. Noticeably, the optical retardance of the electrofabricated membranes was higher than the flow-assembled ones. Presumably, there was momentum carried by the migrating chitosan chains under the electric field that plunged onto the membrane growth front, which likely contributed to the high level of microalignment in electrofabricated membranes, as indicated by the high degree of optical retardance in Figure 6. In the flow assembly case, the shear stress induced by the chitosan flow in parallel to the membrane growth front presumably did not apply extra momentum onto the membrane, which is different from the electrofabrication case. As a result, the quantified optical retardance of a flow-assembled chitosan membrane was in general lower than that of an electrofabricated membrane, as shown in Figure 6.

In future work, the programmable permeability of electrofabricated membranes can be used to create multilayered membranes by varying the applied voltage during formation for selective transport, sorting, or filtration applications [38,68]. We hypothesize that the degradation of the electrofabricated membranes would also change with respect to the fabrication conditions, which we did not investigate in this study. This would provide a promising platform for stimuli-responsive drug release studies, as suggested in the literature [69]. Further, in the current study, we were not able to investigate the mechanical properties (e.g., tensile strength, stiffness) of electrofabricated membranes due to their tiny size and the lack of a sufficient tool for characterization in microfluidics. Future studies can explore this by fabricating macro-sized membranes with a flexible net [35]. To explore the utility of this interfacial electrofabrication for conventional cell or tissue culture studies, macro-sized membranes are also preferred [70].

## 5. Conclusions

In summary, the interfacial electrofabrication of chitosan membranes is a simple, programmable, and robust approach to integrating freestanding biopolymer membranes into a single-layer PDMS microfluidic network. Herein, we demonstrate the programmable physical properties of the resulting membranes by controlling the electrofabrication conditions, which allows for the active manipulation of membrane properties for broader applications. Further, we demonstrate that the electrofabrication of freestanding chitosan membranes can be easily scaled up with high reproducibility and controllability. The ability to simultaneously manufacture membrane arrays and reliably control the membrane growth and physicochemical properties is significant in expanding the use of electrical stimuli for manufacturing and broadening its utility in further studies.

## Figures and Tables

**Figure 1 membranes-13-00294-f001:**
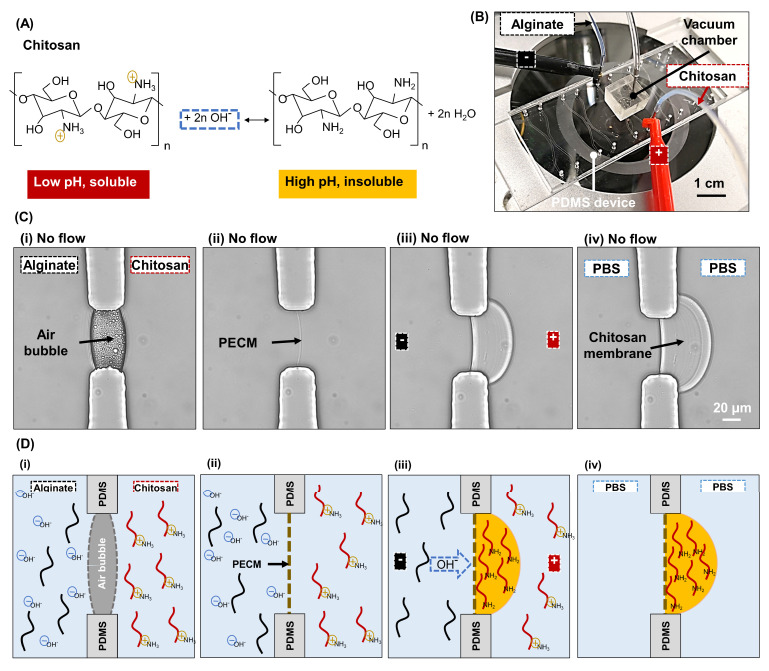
Interfacial electrofabrication of chitosan membranes in a single-layer, single-aperture PDMS microdevice. (**A**) Gelation of chitosan due to interactions with hydroxyl ions. (**B**) Experimental setup for the electrofabrication of chitosan membrane, consisting of PDMS microdevice with an add-on vacuum chamber on the top and two inputs for chitosan and alginate solutions connected to anode and cathode, respectively. (**C**) Microscopic images and (**D**) schematic of chitosan membrane formation process: (**i**) an air bubble was trapped within the aperture; (**ii**) a thin PECM instantaneously formed between the negatively charged alginate and positively charged chitosan macromolecules; (**iii**) a chitosan membrane grown on the PECM due to a flux of hydroxyl ions from the alginate solution driven by the applied electrical potential; (**iv**) the resulting chitosan membranes stored in phosphate-buffered saline (PBS).

**Figure 2 membranes-13-00294-f002:**
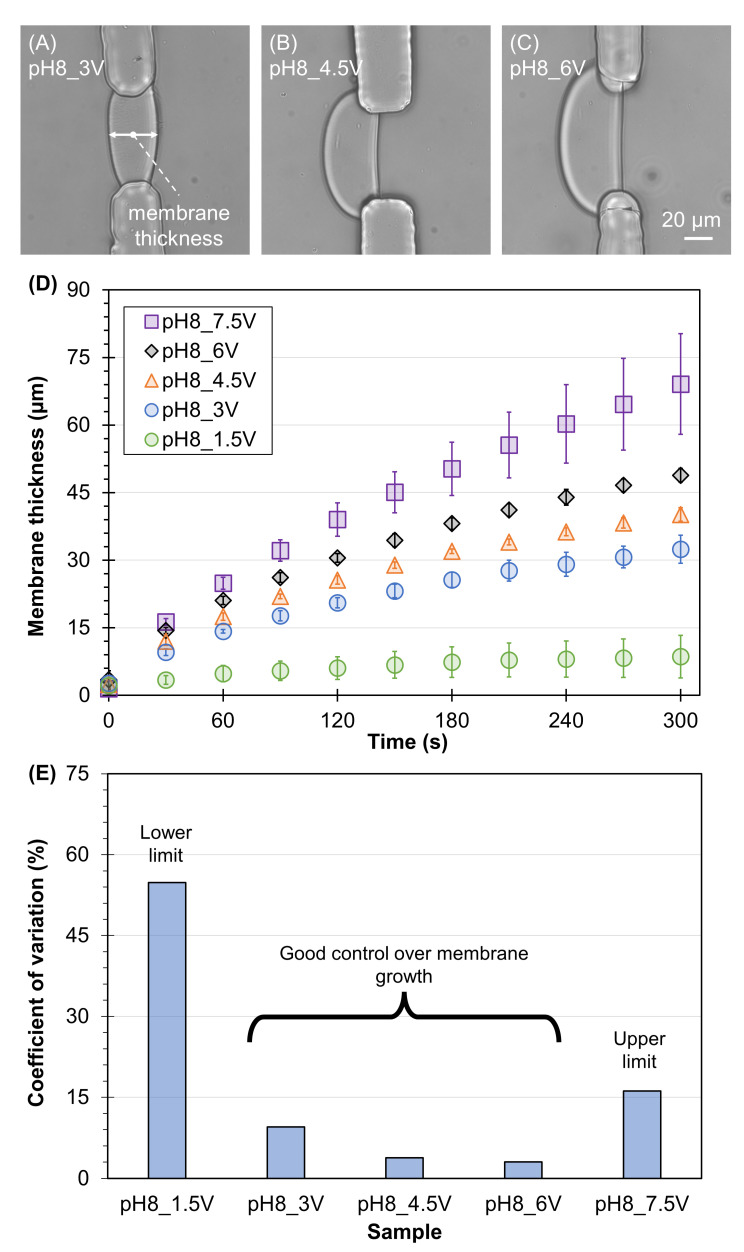
Determining the boundaries of applied voltage for the electrofabrication of freestanding chitosan membranes in microfluidics. (**A**–**C**) Representative chitosan membrane electrofabricated in 5 min using the voltage of 3, 4.5, and 6 V, respectively. (**D**) Typical membrane growth curves of chitosan membranes electrofabricated with different voltages (1.5, 3, 4.5, 6, and 7.5 V) while the pH of alginate solution was fixed at 8. (**E**) The percentages of the coefficient of variation of membranes electrofabricated under varied voltage for 5 min.

**Figure 3 membranes-13-00294-f003:**
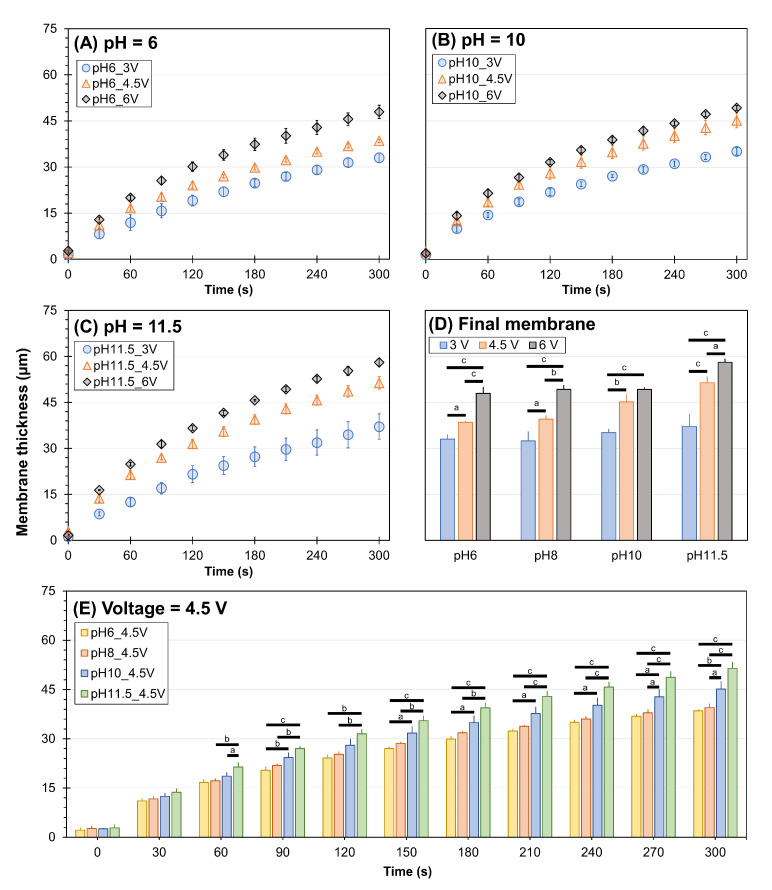
The thickness of chitosan membranes electrofabricated under different conditions. The applied voltage was varied at 3, 4.5, and 6 V while the pH of alginate solution was fixed at (**A**) 6, (**B**) 10, and (**C**) 11.5 (refer to Figure 2 for the case of pH 8). (**D**) Resulting membrane thickness at 5 min at various pH and voltages. (**E**) The applied voltage was fixed at 4.5 V while the pH of the alginate solution was varied at 6, 8, 10, and 11.5. Results of pairwise comparisons are indicated by a (*p* < 0.05), b (*p* < 0.01), and c (*p* < 0.001), with Tukey test comparisons indicated by horizontal bars connecting the compared groups.

**Figure 4 membranes-13-00294-f004:**
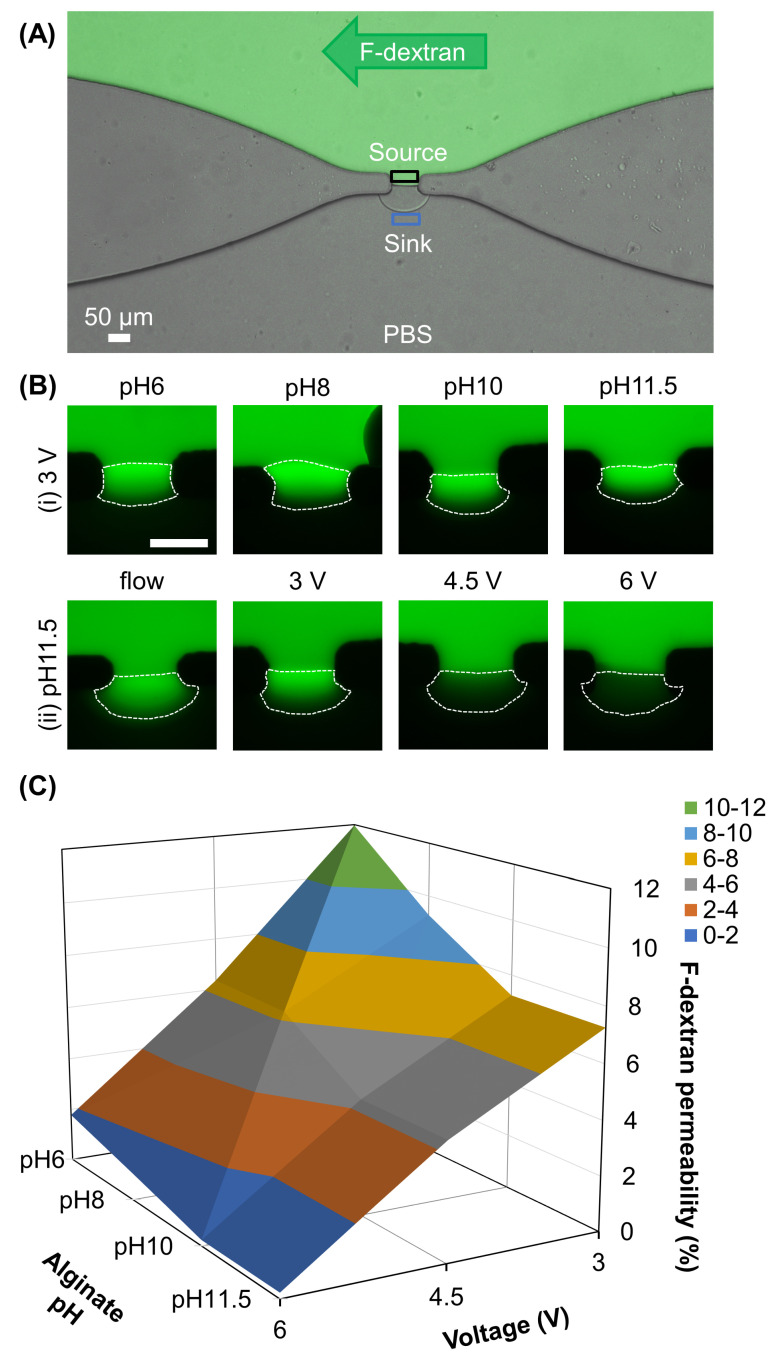
Permeability tests of chitosan membranes. (**A**) Experimental setup of permeability tests. Fluorescence-labeled dextran (F-dextran) was continuously introduced to the upper channel while the lower channel was filled with PBS for 30 min. The percentage of F-dextran that passed through the chitosan membrane was determined by dividing the fluorescence intensity within the blue box (sink) by the fluorescence intensity within the black box (source). (**B**) The zoomed-in images of chitosan membranes, outlined with white dashed lines, under permeability with F-dextran. Chitosan membranes were fabricated at (**i**) fixed voltage (3 V) while varying the pH of alginate or (**ii**) fixed pH of alginate of 11.5 while changing the voltage or fabrication method. (**C**) The permeable percentage of F-dextran across chitosan membranes fabricated under different conditions.

**Figure 5 membranes-13-00294-f005:**
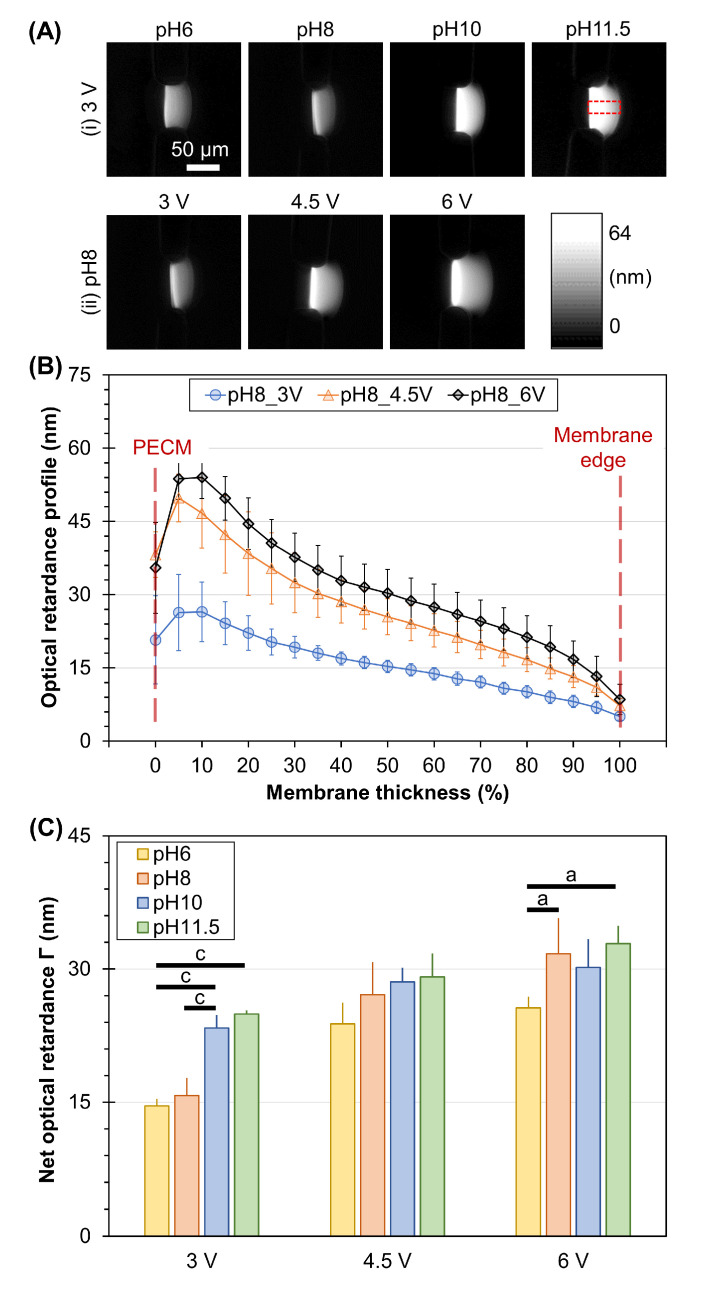
Optical retardance of electrofabricated chitosan membranes. (**A**) Representative optical retardance map of chitosan membranes electrofabricated using (**i**) fixed voltage at 3 V while varying pH of alginate solution and (**ii**) fixed pH of alginate solution at 8 while varying the applied voltage. (**B**) Profile plots of optical retardance within chitosan membrane with respect to the percentage of membrane thickness. (**C**) Net optical retardance of chitosan membranes electrofabricated under varying electrofabrication conditions. Results of pairwise comparisons are indicated by a (*p* < 0.05) and c (*p* < 0.001), with Tukey test comparisons indicated by horizontal bars connecting the compared groups.

**Figure 6 membranes-13-00294-f006:**
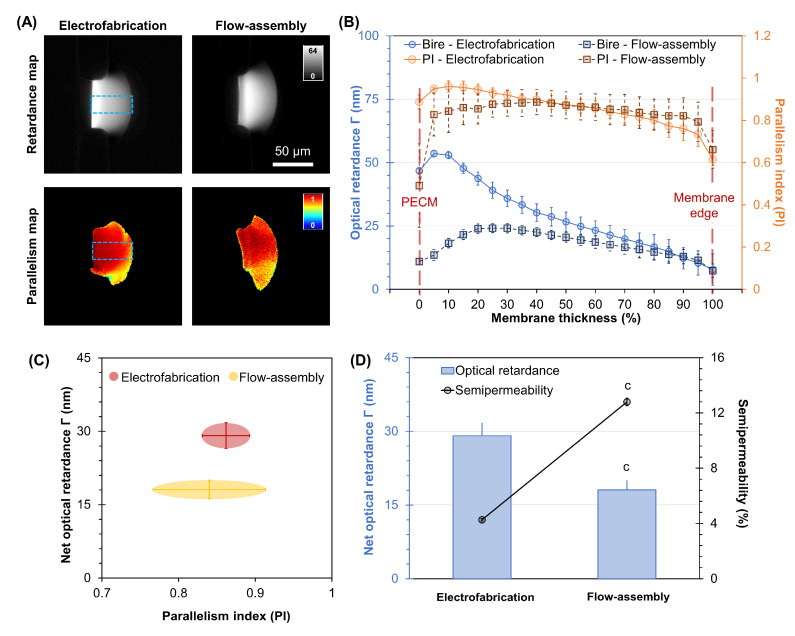
Optical properties comparison between electrofabricated and flow-assembled chitosan membranes. (**A**) Retardance and parallelism index (PI) maps of electrofabricated and flow-assembled chitosan membranes. (**B**) Profile plots of optical retardance and PI across chitosan membranes formed with electrofabrication and flow assembly. Correlations between net optical retardance with (**C**) PI and (**D**) semi-permeability of both types of chitosan membranes. Results of pairwise comparisons are indicated by c (*p* < 0.001).

**Figure 7 membranes-13-00294-f007:**
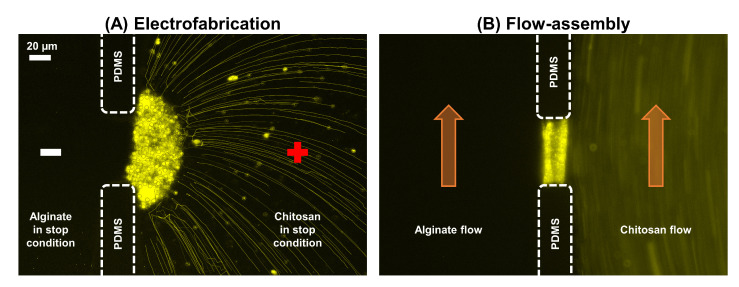
Growth mechanisms of chitosan membranes in the two fabrication approaches. (**A**) In electrofabrication, chitosan chains (anode side) were attracted towards the PECM in the vertical direction, followed by gelation with the flux of hydroxyl ions from the alginate (cathode) side driven by electrical stimuli. (**B**) In flow assembly, chitosan chains were assembled layer by layer along the flow direction and in parallel with the PECM, where the polymerization was induced by the diffusion of hydroxyl ions from the alginate side driven by laminar flows.

**Figure 8 membranes-13-00294-f008:**
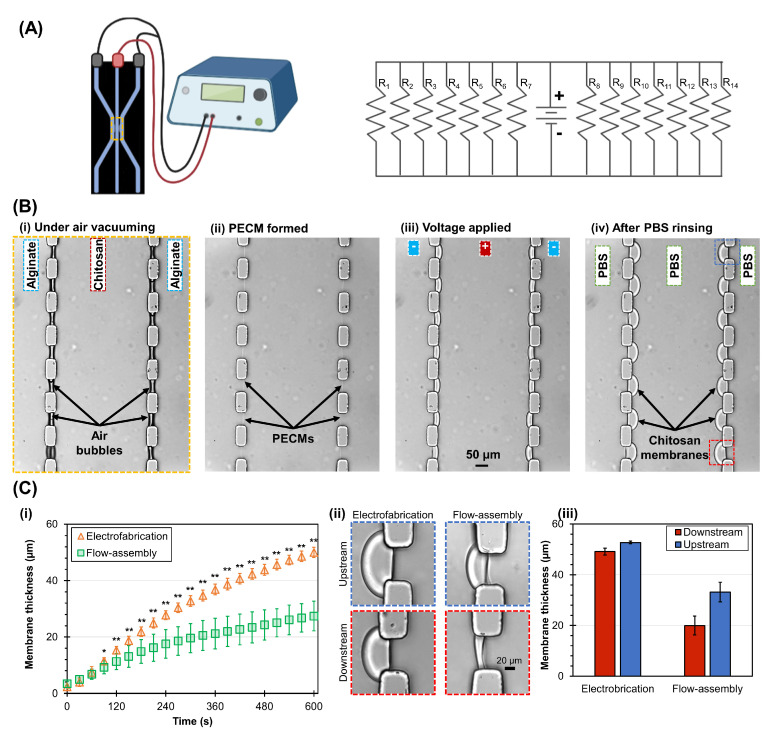
Scaling up the electrofabrication of chitosan membrane arrays in a three-channel PDMS microdevice with distal electrodes. (**A**) Electrical circuit representation of the scaled-up electrofabrication and experimental setup schematics. (**B**) Electrofabrication of chitosan membrane arrays in a three-channel PDMS device in sequence: (**i**) air bubbles were trapped within the apertures and vacuumed d out with an add-on vacuum chamber; (**ii**) PECMs were formed at the solution interfaces upon the contact of chitosan and alginate solutions; (**iii**) voltage was applied to initiate the formation of chitosan membrane arrays in stop flow condition; (**iv**) final chitosan membrane arrays formed and stored in PBS. (**C**) The growth of electrofabricated and flow-assembled chitosan membranes over time: (**i**) typical membrane growth within 10 min; (**ii**) zoomed-in images and (**iii**) membrane thickness of upstream and downstream membranes by the two fabrication approaches. Results of pairwise comparisons are indicated by * (*p* < 0.01) and ** (*p* < 0.001).

**Table 1 membranes-13-00294-t001:** Sample abbreviations with respect to the fabrication parameters.

Sample	pH of Chitosan	pH of Alginate	Voltage (V)
pH6_3V	5.5	6	3
pH6_4.5V	5.5	6	4.5
pH6_6V	5.5	6	6
pH8_3V	5.5	8	3
pH8_4.5V	5.5	8	4.5
pH8_6V	5.5	8	6
pH10_3V	5.5	10	3
pH10_4.5V	5.5	10	4.5
pH10_6V	5.5	10	6
pH11.5_3V	5.5	11.5	3
pH11.5_4.5V	5.5	11.5	4.5
pH11.5_6V	5.5	11.5	6
pH11.5_flow	5.5	11.5	0

**Table 2 membranes-13-00294-t002:** Summary of multiple linear regression analysis of the electrofabricated chitosan membrane properties (e.g., thickness, semi-permeability, and retardance) as functions of pH and voltage. The table summarizes the linear regression coefficients of specific membrane properties in relation to the alginate pH and voltage with coefficients expressed as unstandardized coefficient ± standard error.

Multiple Linear Regression Parameter	Thickness	Semi-Permeability	Retardance
Constant	0.07 ± 0.07	1.83 ± 0.11	−0.07 ± 0.11
Regression coefficient of pH (unit: 1/unit pH)	0.03 ± 0.01	−0.04 ± 0.01	0.04 ± 0.01
Regression coefficient of voltage (unit: 1/volts)	0.1 ± 0.01	−0.23 ± 0.02	0.11 ± 0.02
Multiple R^2^	0.959	0.981	0.935

## Data Availability

The data presented in this study are available in this article, or Appendix A.

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
