# Peer review of "Programmable Physical Properties of Freestanding Chitosan Membranes Electrofabricated in Microfluidics"

_membranes, 2023, doi:10.3390/membranes13030294_

Round 1
Reviewer 1 Report
All comments are provided within the manuscript text (see attachment) - marked with red color.
Moreover: in the discussion, please refer more to the other systems. What are the advantages and disadvantages of the proposed system? Is there any literature on chitosan membranes in similar systems? - please compare your results with these data.

Author Response
Please see the attached response letter. Thank you for your time in reviewing the manuscript and valuable comments to improve the paper.

Reviewer 2 Report
The manuscript describe an interesting approach for the fabrication of chitosan membranes directly integrated in microfluidic devices. The work is well written and clear. I have just some points that need to be addressed:
1. applications should be better discussed (see for instance https://www.sciencedirect.com/science/article/abs/pii/S0019452221000170).
2. in the case of cell culture application, I think that 2 aspects should be addressed: - Have the authors investigated long term stability (over 1-2 weeks) of the membranes at 37°C? - Is it possible to coat the membrane with gelatin, collagen or other adhesive material or these will affect membrane properties/stability?
3. an evident aspect on my opinion is that the membrane has not an uniform thickness. How this can be addressed?
4. in addition to control physical properties, discussing the possibility of tuning mechanical properties can be useful to direct cell behaviour (see for instance https://www.mdpi.com/2306-5354/8/8/106) .
Author Response

(The authors gave the same response as above.)

Round 2
Reviewer 2 Report
The author mainly addresses all the previous points. Just the the last one could be better adrresses:
'in addition to control physical properties, discussing the possibility of tuning mechanical properties can be useful to direct cell behaviour (see for instance https://www.mdpi.com/2306-5354/8/8/106)'
Even if the author do not investigated mechanical properties, I believe this point can be better addressed in the light of cell culture applications, expecially considering the possibility of providing substrates with time evolving properties. Possessing ammine groups chitosan is compatible with enzymatic crossliking strategies replicating physiopathological processes invitro as reported in the suggested ref.
In addition this sentence seems to be in contrast with the application indicated in the intro: 'to explore the utility of this interfacial electrofabrication for conventional cell or tissue culture studies, macro-sized membranes is also preferred'. Microscale tissue culture is widely used for hightroughtput studies. I believe the authors can discuss the possibility of upscaling the technogoly for more flexibility.